# Violins Unveiled: A Photogrammetric Framework Integrating Multiband and Spectroscopic Data for In-Depth Examination of Two Musical Instruments

**DOI:** 10.3390/s25113278

**Published:** 2025-05-23

**Authors:** Federico Di Iorio, Giacomo Fiocco, Riccardo Angeloni, Leila Es Sebar, Sara Croci, Fausto Cacciatori, Marco Malagodi, Federica Pozzi, Sabrina Grassini

**Affiliations:** 1Department of Applied Science and Technology, Politecnico di Torino, Corso Duca degli Abruzzi 24, 10129 Torino, Italy; leila.essebar@polito.it (L.E.S.); sara.croci@polito.it (S.C.); sabrina.grassini@polito.it (S.G.); 2Centro per la Conservazione ed il Restauro dei Beni Culturali “La Venaria Reale”, Via XX Settembre 18, 10078 Venaria Reale (Turin), Italy; federica.pozzi@ccrvenaria.it; 3Arvedi Laboratory of Non-Invasive Diagnostics, Department of Musicology and Cultural Heritage, University of Pavia, Via Bell’Aspa 3, 26100 Cremona, Italy; giacomo.fiocco@unipv.it (G.F.); marco.malagodi@unipv.it (M.M.); 4Museo del Violino, Piazza Guglielmo Marconi 5, 26100 Cremona, Italy; curator@museodelviolino.org (R.A.); cacciatorifausto@gmail.com (F.C.)

**Keywords:** cultural heritage, conservation, diagnostics, non-invasive, photogrammetry, multiband imaging, spectroscopic analysis, physically based rendering, Sketchfab

## Abstract

In the field of cultural heritage (CH), non-invasive analyses, such as photogrammetry and multiband imaging (MBI), play a pivotal role as effective solutions for examining the morphology, materials, and state of preservation of an artifact. Gathering such information is particularly valuable since these data are complementary and provide a comprehensive perspective for an in-depth study of a wide variety of historically and artistically significant artifacts. Photogrammetry and MBI are commonly utilized for these purposes but typically as separate methodologies. This research seeks to address this limitation by integrating these datasets to enrich the information embedded within a 3D model, thereby facilitating the identification of areas subsequently analyzed using spectroscopic techniques. This study provides an in-depth analysis of two historically significant violins housed at the Museo del Violino in Cremona (Italy) contributing to a more comprehensive understanding of a specific category of artifacts that remains underrepresented in the existing literature. Furthermore, the technical workflow for integrating MBI data using the Physically Based Rendering (PBR) approach and Sketchfab, along with the interpretation of the resulting data, is presented.

## 1. Introduction

### 1.1. Fundamentals and Advantages of Photogrammetry in Cultural Heritage

Photogrammetry is a technique that enables the creation of three-dimensional models from photographic images. This methodology, now widely adopted in the cultural heritage (CH) field and continuously expanding, is used to reproduce objects and environments of various types and materials [1,2,3,4,5,6,7,8,9]. The ability to produce accurate and cost-effective 3D models represents the primary advantage of this approach when compared to more expensive methods such as LiDAR (Light Detection and Ranging), SLAM (Simultaneous Localization and Mapping), and structured light scanning. Due to its ease of use, photogrammetry has progressively been employed over the years for the large-scale and systematic digitization of museum collections [10] and for performing monitoring campaigns aimed at planning restoration interventions [11,12].

The main advantages of using photogrammetry are (i) the ability to perceive a subject in its entirety; (ii) performing metric computations (calculation of areas, volumes, and linear distances); (iii) deriving sections, plans, and profiles; (iv) producing high-resolution two-dimensional undistorted views (orthomosaics); and (v) the ability to compare different models of the same subject over time to monitor potential deviations. Furthermore, photogrammetry is widely employed in the generation of additional outputs that enhance the understanding of the reproduced subjects. One example is represented by V-RTI (Virtual Reflectance Transformation Imaging), a hybrid technique that combines the properties of RTI with photogrammetry, enabling the extraction of surface data from a polygonal mesh [13,14]. Hybridizations with other methodologies, for which photogrammetry serves as an essential data foundation, include the capability to transform surfaces with curved spatial coordinates into planar coordinates to achieve an optimal readability [15,16,17]; aggregate photogrammetric data with laser scans, total stations and GNSS for the combined survey of complex sites [18,19]; provide data for subsequent use in CAD (Computer-Aided Design) environments [20], with further systematization within GIS (Geographical Information System) [21] and HBIM (Heritage Building Information Modelling) outputs for segmentation and risk analysis [22]; semantically characterize architectural elements through supervised and unsupervised machine learning strategies [23,24]; integrate with thermal data for an in-depth analysis of the internal structures of buildings [25,26]; and serve as a starting point for Principal Component Analyses (PCAs) aimed at studying the materials and historical background of artifacts [23,27].

### 1.2. Integration Challenges of Multiband Data in 3D Models

Despite the numerous advantages of integrating photogrammetry with other techniques, difficulties in aggregating multiband data (MBI) have hindered its widespread use in CH diagnostics. Numerous solutions have been developed over time to address specific sharing needs on various online platforms [28,29,30,31,32,33,34,35,36,37], but MBI data are often incomplete or challenging for non-specialized users to incorporate, making real-time comparisons less immediate. While in the 2D domain it is standard practice to produce multilayer files that allow for a more immediate comparison between the various bands, this approach seems to be less widespread in the 3D field, and the methodologies applied less similar to each other, mainly opting for the production of a different model for each spectral range.

Despite the limitations that preclude the simultaneous use of the available data, multiband photogrammetry (MBI 3D) is rapidly spreading in the conservation and restoration field [38,39,40]. This is due to its ease of use, the optimization of photogrammetry software, and the relative inexpensiveness of the required instrumentation. The scientific literature includes numerous applications in this field, ranging from CH to remote sensing techniques for environmental and architectural monitoring. Various techniques are employed in different ways to address a wide variety of specific needs [27,41,42,43,44,45,46,47,48,49]. This heterogeneity of approaches suggests that no one method is better than another but rather a complex system of solutions designed from time to time to cope with the different dimensions and materials that objects of historical and artistic interest are characterized by.

In the field of scientific research applied to musical instruments (e.g., archeomusicology), several studies have employed digital techniques—such as 3D scanning and printing or computed tomography—to obtain valuable information on the manufacturing process, to assess the conservation state, and to investigate the shape–sound relationship [50,51]. However, non-invasive methods such as multiband imaging, as well as challenges related to real-time comparison and data dissemination, are often not addressed comprehensively.

### 1.3. Multiband Photogrammetry and Spectroscopic Techniques for Historical Violin Analysis

MBI and spectroscopic analysis offer a non-destructive means of studying materials by exploiting their characteristic interactions—reflection, absorption, and emission—with electromagnetic waves, which vary with the molecular composition and structural properties. Using a range of spectral bands and energy levels, each technique yields specific insights into the artwork. For example, the luminescence technique is commonly applied in the heritage field to distinguish between different varnish applications and retouchings, whereas near-infrared reflectography is used to penetrate the surface to uncover hidden underdrawings and the artist’s pentimenti. The application of MBI to historic musical instruments—where non-invasive methods are strongly preferred—offers considerable promise. In bowed instruments, for instance, mapping the distribution of past conservation treatments, characterizing varnish layers, and detecting structural features such as woodworm galleries, cracks or surface deposits invisible to the human eye, can be critical for assessing the conservation state of an object of such artistic relevance [43,52,53,54]. In this study, we employ several MBI techniques, including ultraviolet-induced luminescence and infrared reflectography, to pinpoint interesting areas for detailed spectroscopic analysis.

Historical stringed instruments have been the object of great scientific interest for many years, in addition to their historical and construction value. The study of the construction features, as well as the surface finishing treatments and all the materials used in the varnishing stages by the most important historical Cremonese violin makers, is crucial for deepening our understanding of the manufacturing process and preserves the long tradition of historical Cremonese violin making, which is still today at the forefront of this particular artistic craft [55].

For this reason, the development of a methodology to integrate spectroscopic results with imaging and three-dimensional information is crucial in the study of this particular type of artwork. As widely documented in the scientific literature [52,53,56,57,58], the study of varnishes and finishing materials on historical violins requires, in most cases, non-invasive techniques that do not damage the surfaces of these precious stringed instruments. In this sense, the use of External Reflection Infrared spectroscopy perfectly intercepts the needs of the study, allowing rapid access to information on organic materials (e.g., siccative oils, resins, adhesives, and sealers) and inorganic materials (fillers, ground, and pigments).

As a result of these needs, this study aims to develop a methodology to (i) rapidly and comprehensively document valuable, medium-sized unique objects (e.g., violins); (ii) integrate multiband imaging into a single 3D model supporting a simultaneous and a more efficient comparison; (iii) identify Regions of Interest (ROIs) for further spectroscopic analysis; (iv) validate the photogrammetric model by comparison with data from a structured-light scanner, and (v) use a user-friendly web platform such as Sketchfab adapting the Physically Based Rendering (PBR) control-panel layout to share the results.

Therefore, the next sections propose a description of the equipment to collect spectral data, the methodology to integrate them in the 3D models, and the results of the non-invasive analyses performed on two musical instruments from the Museo del Violino di Cremona (Italy).

## 2. Materials and Methods

This section will describe the case studies, the methodology, and the tools for acquiring multiband and spectroscopic datasets, along with the procedures for their elaboration. The steps for integrating the MBI 3D results on the Sketchfab, already widely used in CH for several research purposes [59,60,61,62,63,64], are then described.

### 2.1. Historical Violins from the Museo Del Violino of Cremona

The study focuses on two historical violins from the Museo del Violino of Cremona collections (Figure 1).

The first one is an anonymous violin recently attributed to the circle of the Cremonese-trained, Brescian-based maker, Giovanni Battista Rogeri (c. 1642–c. 1710), inventory number S. 107 (referred to in this manuscript as Inv. 107) (Figure 1a,b). The second one is the “Carlo IX” made by Andrea Amati around 1570, inventory number S. 81 (referred to in this manuscript as Inv. 81) (Figure 1c,d). It is considered a masterpiece of the renowned set of early violins allegedly produced by Amati between 1566 and 1572 for the court of Charles IX of France (1550–1574) [65].

Italian instruments of the violin family, especially the ones made between the 16th and 18th centuries, have been appreciated for their aesthetic and tonal features and collected, prized, and studied throughout recent centuries. Bowed and plucked string instrument makers were already working in Brescia as early as the end of the 15th century [66]. As for Cremona, Andrea Amati (c. 1505–1570) is generally recognized as the forefather of the Cremonese violin-making tradition. He developed a design and construction method that relied on an internal mold, a key innovation that shaped the long-lasting success of his students, followers, and those who continued his work [67]. The “Carlo IX” is one of the earliest examples of this tradition, and like all the instruments of the French royalty set, it features painted and gilded decorations. In this case, there is a partially readable motto on the ribs (PIETATE ET IUSTITIA) as well as allegorical figures, ornaments, and coats of arms on the back and the scroll [68].

### 2.2. Analytical Methods

#### 2.2.1. Multiband Photogrammetry

Data are recorded according to the theoretical workflow depicted in Figure 2.

The MBI 3D is recorded in the following spectral ranges: Ultraviolet Reflectography (UVR), Visible Light (VIS), Infrared Reflectography (IRR), and Ultraviolet Luminescence (UVL). Additionally, two false-color images derived from IRR and UVR are generated during post-processing, namely IRFC and UVFC.

For MBI 3D, a modified camera, specifically a Fujifilm X–T30 (Fujifilm Holdings Corporation, Tokyo, Japan), is employed. The choice of this type of camera over a dedicated MBI camera is justified by its higher resolution and lower cost, compared to dedicated MBI cameras, which offer limited resolution. Additionally, many of them have a constrained spectral range that does not support the simultaneous capture of near-ultraviolet and near-infrared wavelengths with the same device. The Fujifilm X–T30 employs a X–Trans sensor, which is based on silicon. The theoretical photosensitivity of this sensor is determined by the spectral response of silicon [69] and is expected to range from 200 to 1100 nm [70]. In practice, this range varies depending on the specific sensor model and, consequently, on the camera. Standard cameras are equipped with an infrared cut-off filter placed in front of the sensor. In a modified camera, such as the one employed in this study, the infrared filter is removed and replaced with one that enables full-spectrum capture.

The pictures are taken in the absence of ambient light and in a room with stable humidity and temperature to limit the thermo-hygrometric stress on the two violins. The camera is placed on a tripod, and the lights are fixed to two stands on either side of the camera position. The background, the workplace, and the walls of the room are covered with dark material to contain any color cast on the subjects, and to limit any artifacts when texturing the models.

The violins are photographed with the use of a turntable, and the shooting scheme involves four revolutions around each object, recording an image approximately every 10° of rotation to ensure a good image redundancy. Each picture is acquired with the aid of a remote control to avoid any vibration during the exposures. Two revolutions are performed at different heights with the objects in a vertical position, and two more by placing the violins firstly facing up and then facing down. For each shooting position, four bands are recorded in the sequence UVR, UVL, VIS, and IRR (and vice versa), in an order established to optimize the change of filters, lights, and focus as depicted in Figure 3.

For each model, the following datasets are obtained:504 images (126 × 4 bands) for Inv. 107;476 images (119 × 4 bands) for Inv. 81.

Table 1 describes the technical details for recording the different spectral ranges.

The graphs depicted in Figure 4 show the filter transmittance and the spectral emission of the radiation sources used for multiband analysis.

As illustrated in Table 2, the images are captured whilst maintaining constant aperture and ISO values, adjusting the exposure time and focus when required. In order to enhance the efficiency of image acquisition, particularly for the UVR and UVL sessions, a slightly higher sensitivity value is adopted with the aim of reducing the acquisition times. The increase in digital noise observed at 200 ISO is found to be negligible when compared to that observed at the native sensitivity of the Fuji X–T30 (160 ISO). The images are captured in RAW format and then converted to JPG after minimal processing in Adobe Photoshop Camera Raw (v.24). Adjustments include exposure correction, color calibration, white balance tuning, reducing the global contrast, and adhering to data conservation principles to preserve the original integrity of the files. For the UVR and IRR, namely, the blue and red channels are selected, as they represent the lowest noise information for each band [48]. The color calibration for the VIS dataset is processed with Calibrite^®^’s ColorChecker Camera Calibration software (v. 2.2.0) using the Calibrite^®^ ColorChecker Passport (Calibrite LCC, Wilmington, DE, USA) colorimetric target, while the UVL one is balanced using UV Innovation’s Target–UV™ [71,72,73].

#### 2.2.2. 3D Data Processing

In order to dress the 3D models with the various textures, the images collected at different bands must be perfectly superimposed on top of each other. Photogrammetry software may produce errors in registering images from different spectral ranges. Therefore, before starting the 3D reconstruction process, it is necessary to register the individual shots on the basis of the VIS dataset, ensuring a perfect match of the images of the entire MBI dataset, which are usually affected by a change in scale in each range. This phase is known to be particularly time-consuming due to the large number of shots involved, especially in the absence of an apochromatic lens [74]. The algorithm available in the version of Metashape used in this study proves to be ineffective for multiband data registration, yielding inaccurate results. Algorithms such as Maximization of Mutual Information (MMI) and Speeded-Up Robust Features (SURFs) have been developed over the years in various research domains to reduce processing time and minimize errors associated with manual registration [40]. Despite the high level of accuracy offered by these systems, these methods are not efficient in terms of computational resources, particularly when dealing with high-resolution images and large datasets typical of artwork documentation. Given the considerable size of the datasets and a greater familiarity with more traditional methods, the authors chose a manual procedure that enabled to directly evaluate the quality of the process image by image, although this approach may result in a longer processing time. It is important to underscore that, once the central pixel of the images was aligned, the only transformation applied was a scaling adjustment, avoiding non-linear deformations and arbitrary modifications to preserve the geometric integrity of each acquisition. This phase can be accelerated by minimizing system vibrations and optimizing filter changes and focus adjustments (Figure 3). In a hypothetically static system, multiband images at different wavelengths differ solely due to a radial scale variation originating from the optical axis.

After ensuring the registration of the central pixel across the different bands, correction of this distortion requires the application of an identical scaling factor (to be determined relative to the reference VIS image) for all images within the same band. Once the files are organized into their respective sub-folders, they are renamed using sequential numbers, each number being assigned to the files of the four bands taken from the same position. Subsequently, the model reconstruction is carried out by importing only the VIS dataset. The MBI-to-VIS registration is a requirement for 3D processing, as in the final stage, the VIS images will be replaced with the other bands to generate different textures on the same model.

The 3D processing is performed with Agisoft Metashape Professional 2.0 [75]. The pre-processing involves the preparation of masks to be applied to the image files. The purpose of masking is to prevent the identification of matching points that do not belong to the subject, which could hinder the reconstruction. For this purpose, the VIS dataset is initially divided into three chunks, each one corresponding to the violins’ positions. Each chunk is used to rapidly create a low-poly model, from which polygons not belonging to the instruments (the background and the rotating platform) are removed. At the end of the process, Metashape is able to extract the masks directly from the 3D model, avoiding the manual masking procedure and significantly speeding up this phase [76]. Thereafter, the images of the second and third chunks, along with their associated masks, are transferred into the first chunk to restart the 3D reconstruction. The MBI 3D workflow can be summarized in the following steps:Image alignment and tie point sparse cloud generation;Filtering the worst point by Gradual selection (removing 1/10 of the tie points from projection accuracy, reprojection error, and reconstruction uncertainty);Camera optimization;Depth maps calculation (depth filtering: aggressive);Meshing (interpolation: enabled);Decimating (between 20k and 50k polygons);VIS texture generation;NM (Normal Map) generation based on the high–resolution model;Export of the VIS model (along with NM) in OBJ format;VIS images substitution with the MBI bands;Re-texturization (keep uv);Export of the MBI texture.

To obtain a high-resolution model, it is sufficient to skip steps n° 6 and 8. The NM is later useful in Sketchfab (or other 3D viewers) to simulate the high-res detail on a low-poly mesh. The model is decimated to reduce file size and to enable a smoother online visualization. A high-res mesh will be used separately for detailed morphological analyses (surface computations, cross-section extraction, etc.). This study focuses on producing an optimized model for displaying multiband textures and to assess the conservation state of surface materials. Polygon reduction aligns with this purpose, allowing multiple textures to be applied to a 3D model regardless of the quantity of polygons.

#### 2.2.3. Morphological Evaluation with a Structured Light Scanner (Artec Eva)

The 3D model of the violin Inv. 107 is scaled in CloudCompare (v2.13) using a high-precision digital replica. This replica, generated via an Artec Eva structured light scanner (accuracy: 0.1 mm, resolution: 0.2 mm), serves as the metrological reference to ensure dimensional accuracy during the scaling process. The alignment of the two meshes is performed using the Iterative Closest Point (ICP) algorithm, with a scale factor adjustment applied within the coordinate system of the 3D scanner model, which serves as the metrological ground truth. Morphological discrepancies are quantified by calculating the RMSE (Root Mean Square Error) and the Standard Deviation.

#### 2.2.4. Physically Based Rendering and Sketchfab

The integration of multiband textures is carried out using Sketchfab. This online rendering engine does not natively allow the upload of additional textures beyond the visible one, and thus its functionality has to be adapted to meet the needs of this study. The control panel for 3D attributes provided by this web viewer is similar to those found in most rendering software developed for modeling and animation (e.g., Blender [77]). To simulate the behavior of light on different materials, these engines rely on the PBR approach [78,79,80], which manages how light interacts with objects through the control of so-called material shaders. Among the most common types are specularity, opacity, roughness, glossiness, metalness, and anisotropy; each of these defines different qualities that rely on complex computational algorithms to be reproduced. For many years, there has been a growing interest in understanding PBR for the creation of high-quality 3D models, leading to an increasingly conscious use of this approach even in the documentation of cultural heritage [81,82,83]. However, in this study, material shaders are exclusively employed to host the various multiband textures, without assigning the model the characteristics for which they were originally designed. The data are arranged as follows:VIS: Base Color;UVL: Anisotropy (value: 0.001);IRR: Cavity (value: 0.001);UVR: Clear Coat (value: 0.001);UVFC: Sheen (value: 0.001);IRFC: Emission (value: 0.01);NM: Normal Map.

The seven textures generated through and after the photogrammetric process (VIS, UVL, IRR, UVR, UVFC, IRFC, and NM) are assigned to the material shaders in Sketchfab’s PBR editing panel. The assignment of minimal values to each shader guarantees the inclusion of the files in their designated layers without altering the final appearance of the model. Because several attributes accept only greyscale data, color textures (VIS, UVL, UVFC, IRFC, and NM) are allocated to the RGB-compatible ones—Base Color, Anisotropy, Sheen, Emission, and Normal Map—while the monochromatic reflectographies (UVR and IRR) are assigned to the Clear Coat and Cavity channels. Upon integration, these MBI textures can be selectively activated or deactivated within Sketchfab’s Model Inspector during the visualization, allowing real-time comparative analysis of the spectral layers.

#### 2.2.5. Spectroscopic Analysis

External Reflection FTIR (ER-FTIR) spectroscopy is conducted using the Alpha portable spectrometer (Bruker Optics, Billerica, MA, USA) equipped with the R-Alpha external reflectance module, featuring an optical layout of 23°/23°. The compact optical system includes a SiC Globar source (Bruker Optics, Billerica, MA, USA), a RockSolid interferometer with gold mirrors (Bruker Optics, Billerica, MA, USA), and an uncooled DLaTGS detector (Bruker Optics, Billerica, MA, USA). The reflectance module enables non-contact and non-invasive measurements of areas approximately 5 mm in diameter, with a working distance of 15 mm. Pseudo-absorbance spectra [log(1/R), where R represents reflectance] are recorded over the range of 7500 to 375 cm^−1^, with a resolution of 4 cm^−1^ and an acquisition time of 1 min [84]. A gold flat mirror is used to acquire the background spectrum. For the analysis of derivative bands, reflection infrared spectra are converted into absorbance spectra using the Kramers–Kronig transformation (KKT), available in the OPUS software package version 7.2 (Bruker, Billerica, MA, USA).

## 3. Results and Discussion

### 3.1. Multiband Photogrammetry

As demonstrated in Figure 5e and Figure 6, the final models are presented when the various textures are selected on Sketchfab. The combination of the UVR (Figure 5a) and IRR (Figure 5d) with the VIS (Figure 5c) lead to the production of the two false colors, IRFC (Figure 5e) and UVFC (Figure 5f) [85,86]. This process is replicated for the “Carlo IX”.

The models’ data are listed in Table 3. Image masking and optimization work are performed with Gradual selection (par. 2.2.2), ensuring sub-pixel alignment values (Reprojection error).

The analysis reveals that specific areas of the violins are not captured with enough images, leading to a less accurate reconstruction of some small details (Figure 7).

### 3.2. Metrological and Dimensional Assessment

Regarding Inv. 107, the morphological discrepancies between high-res photogrammetry and 3D scanner are investigated using CloudCompare’s ICP algorithm.In order to perform a rigorous evaluation of the comparison between two models generated by different sensors, it is standard practice to establish Ground Control Points (GCPs) in the acquisition setup. Although these are essential to ensure proper registration and to evaluate the quality of the process, in this case, it is not possible to employ them because the scans are taken at different times, making it impossible to place the instrument in the same reference system. Therefore, in the absence of GCPs, it is necessary to adopt an alternative approach that contemplates the use of the ICP algorithm. The model made with the 3D scanner thus serves a dual function: firstly, it provides the dimensional reference for scaling; secondly, it enables the measurement of morphological discrepancies in the model obtained by photogrammetry. The data presented in Figure 8 reveal that the majority of the values exhibit a Standard Deviation of 0.54 mm from the reference model, with a Gaussian mean of 0.17 mm. Given the purposes of this study, the accuracy can be considered acceptable.

### 3.3. Orthomosaics Projections

The digital replicas prove to be a valuable resource in the production of several MBI orthomosaics (orthographic views generated from the 3D models), yielding analogous data to those obtained through conventional 2D documentation. Once the model is oriented according to its primary axes, it is possible to generate high-resolution views (top, bottom, front, and back) that can be assembled into multilayer files. This allows not only to identify ROIs which reveal features worth investigating with spectroscopic analysis, as primarily intended in this study, but it allows also to reveal interesting phenomena of a conservative nature. Figure 9 and Figure 11 show some close details of the two violins.

### 3.4. Identification of ROIs and Spectroscopic Investigation

#### 3.4.1. Andrea Amati “Carlo IX” (Inv. 81)

Exploration of the 3D models of both violins (Figure 5a–f and Figure 6a–f) presented using different textures allowed us to identify some ROIs to focus spectroscopic investigations for chemical characterization of the possibly original finishing materials. As for Inv. 81, two ROIs are identified (Figure 9) on the back plate: ROI–A at an area with probably original varnish residue and ROI–B, representative of areas no longer painted and with exposed wood. ER–FTIR investigations (Figure 10) show the presence, at ROI–A, of a mixture of siccative oil (e.g., linseed oil) and natural resins (e.g., colophony) [56]. In addition, in some spectra acquired in the same area, there are signals corresponding to shellac [87] and, probably, benzoin resin [53], both non-original materials used during the maintenance and polishing phases of the violin [55]. At ROI–B, where the varnish is no longer present as shown by the MBI 3D investigation, characteristic bands of proteinaceous material [57] attributable to a possible glue-based wood preparation (e.g., animal or casein glue) as documented on other violins of the Cremonese school [53], are highlighted (Figure 10b). The presence of glue as a sealer is associated with the possible identification of characteristic signals of silicates (e.g., talc) [87] (Figure 10b), attributable to dispersed particles within the preparatory layer [88]. However, it cannot be ruled out that the silicates come from materials used during polishing procedures. In the same areas, characteristic signals of shellac and benzoin resin are also identified (Figure 10a). In addition, always on the back of the Carlo IX, the observation of the various textures enables the identification of other ROIs (Figure 9C–F): in ROI–C and –D, the evidence of neck replacement procedures at the button is shown, while in areas E and F, it is possible to observe some details of the allegory of divine mercy [65], almost illegible in Visible Light.

#### 3.4.2. Anonimo Bresciano
(Inv. 107)

For Inv. 107, three ROIs are identified (Figure 11) on the back plate, which shows different UV fluorescence in areas A, B, and C. Observing the distribution and overlapping of the three different materials on the back, it appears that area A may correspond to the lower original varnish, while B and C correspond to different superimposed restoration layers. ER–FTIR investigations conducted at area A (Figure 12a) identify the presence of shellac as an external layer of the finishing treatment, with the possible addition of natural resins [55]. However, some signals might also confirm the presence of an oil-based varnish subsequently covered by shellac (Figure 12a) [87]. The same signals related to these varnishes are also confirmed in ROIs B and C, together with the characteristic bands of silicates (Figure 12b,c). In particular, the bands highlighted in Figure 12b,c can be attributed to quartz and kaolin [89], which are probably related to the presence of iron-based earth pigments (e.g., red ochre and sienna) [89]. In addition, the possible presence of carbonates is also detected in B and C (Figure 12b,c) [87]. At ROI–C, but to a smaller extent also at A and B, typical signals of a long-chain aliphatic compound (e.g., wax) (Figure 12c), probably used during polishing steps, are identified [55]. In-depth observation of all textures makes it possible, also for this violin, to identify some interesting details on the back of the instrument (Figure 11). In particular, through IRR (Figure 11D) and IRFC (Figure 11E), it is possible to highlight the presence of cracks and surface repairs of potential woodworm galleries, while UVR identifies a fingerprint in the center of the back (Figure 11F).

## 4. Conclusions

The integration of multiband imaging (MBI) and photogrammetry represents a promising yet complex area of research, characterized by a wide variety of methodological approaches, each of which has specific advantages and limitations. Currently, available solutions provide a high level of customization and control; however, their functionality is often confined to local systems, restricting the sharing and accessibility of results. While the customized approach may be effective for specific applications, it poses significant challenges for remote collaboration and large-scale implementation. Conversely, the sharing of multiband photogrammetry (MBI 3D) encounters intrinsic difficulties related to the data structure, as a separate model has to be generated for each spectral band, limiting the identification of areas of scientific interest. This limitation prevents integrative analyses, which could instead provide a more complete understanding of the materials.

To overcome this, an approach adapting the Physically Based Rendering (PBR) layout, used in web-based platforms such as Sketchfab, was specifically designed. This method was employed to facilitate a simultaneous comparison of multiband data within a 3D model, thereby enabling the dissemination of the results on the web and extending their accessibility to non-specialized professionals. The intuitive interface of Sketchfab has been commended by community members for its ease of use, which simplifies content uploads for both institutions and individuals and ensures effortless access for viewers. Beyond its technical strengths, Sketchfab has also proven to be valuable in an educational context. This integration enhances the learning process by introducing interactive, immersive ways to engage with cultural heritage, and, such as our proposed methodology, helps in the integration of multiband data. Despite its user-friendly interface, one of the main limitations of our approach is represented by the use of Sketchfab itself. Looking ahead, the real challenge will be its planned dismission in the upcoming months, to be replaced by Fab, a commercial platform for 3D model distribution, launched at the end of 2024, that has given rise to a number of concerns from a variety of cultural institutions [90]. This shift poses a significant barrier for the cultural heritage sector. We are confident, however, that alternative solutions under development within recent European initiatives will come to fruition. Despite being driven by the closure of the largest online community dedicated to 3D heritage sharing, this transition presents a valuable opportunity to establish a dedicated 3D infrastructure for cultural heritage, featuring enhanced tools and sharing licenses (for example, embedded metadata to support inter-operability) tailored specifically to this field. Until then, our methodology remains applicable to any future web platform that, like Sketchfab, supports PBR and material shaders for rendering and visualizing 3D models.

MBI 3D allows numerous outputs to be obtained from each model, including the orthomosaics. The latter was used to document and identify the distribution of the materials on the surface of the violins and to highlight some important details, yielding results comparable to those attainable through traditional 2D documentation. The collected data led to the selection of ROIs to be investigated through ER–FTIR spectroscopies. The two analytical techniques revealed the presence of the most common materials already identified on many historical violins [55], such as oil-resin varnish, natural resins, shellac resin, protein material, carbonates, and silicates. The workflow outlined in this research is one potential strategy for integrating diverse spectral data, representing a starting point for further investigation.

Future research will aim to optimize the simultaneous capture of Visible Light (VIS) and multiband imaging (MBI) datasets. By minimizing redundant data collection, this approach will streamline workflows, particularly for objects with simple geometries such as historical violins. Subsequent studies will evaluate the scalability of this approach in large-scale architectural and archaeological contexts, where the integration of multi-source data—such as GNSS, total stations, laser scanners, photogrammetry, and MBI—holds significant potential for advancing cultural heritage documentation and analysis.

## Figures and Tables

**Figure 1 sensors-25-03278-f001:**
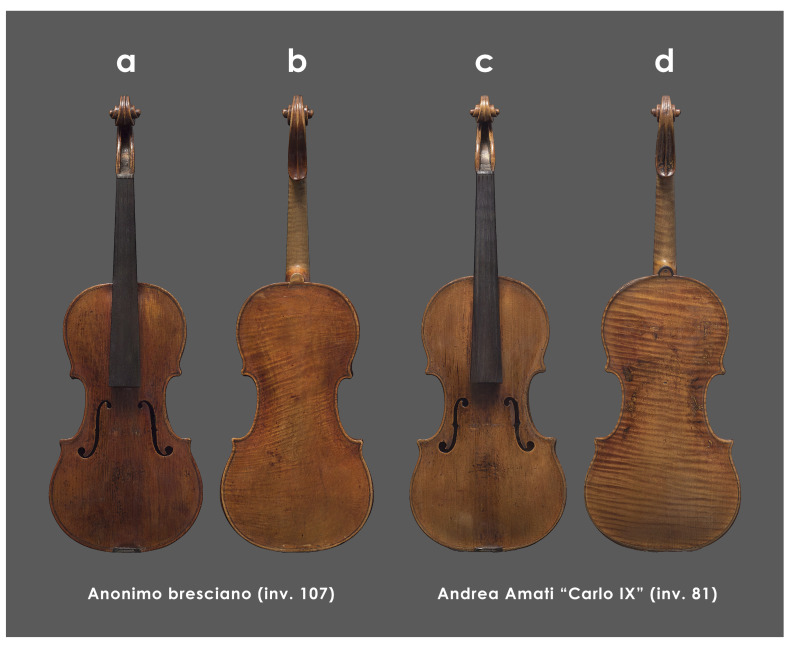
The 3D model overviews in Visible light of the Inv. 107 (**a**,**b**), and of Inv. 81 made by Andrea Amati around 1570 (**c**,**d**).

**Figure 2 sensors-25-03278-f002:**
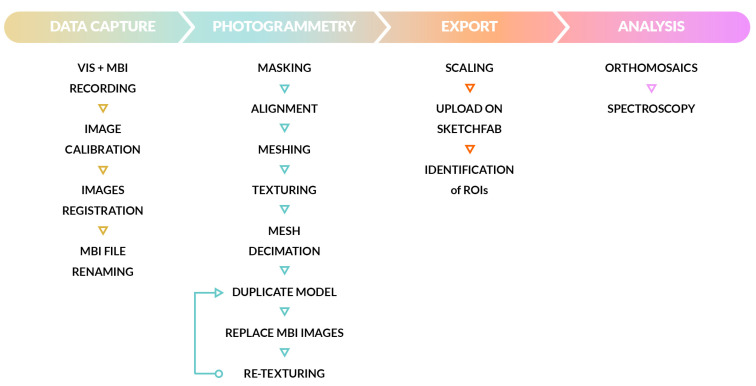
The flowchart describes the main steps of the theoretical workflow for data acquisition, processing and analysis.

**Figure 3 sensors-25-03278-f003:**
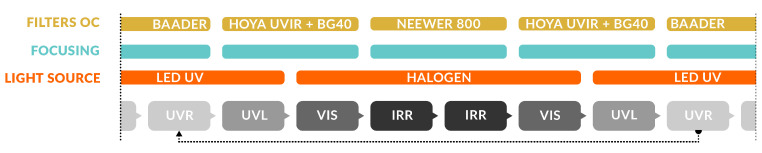
Shooting sequence for MBI 3D. The workflow is optimized to reduce the acquisition time and to limit the system vibrations. ‘Filters OC’ refers to the filters mounted on the camera. With regard to ’Focusing’, since each wavelength falls in a different plane onto the camera sensor, the focus of each multiband image must be adjusted accordingly. The infographic illustrates the correction of the focal plane for each band, based on the use of filters and lamps.

**Figure 4 sensors-25-03278-f004:**
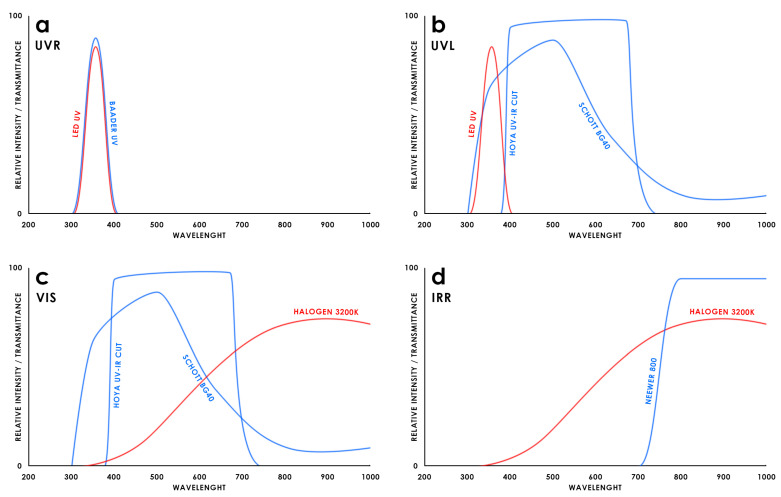
Configuration of filters and lamps used to record the various spectral ranges. (**a**) Ultraviolet Reflectography (UVR); (**b**) Ultraviolet Luminescence (UVL); (**c**) Visible light (VIS); (**d**) Infrared Reflectography (IRR). Filters are shown in blue and lamps in red.

**Figure 5 sensors-25-03278-f005:**
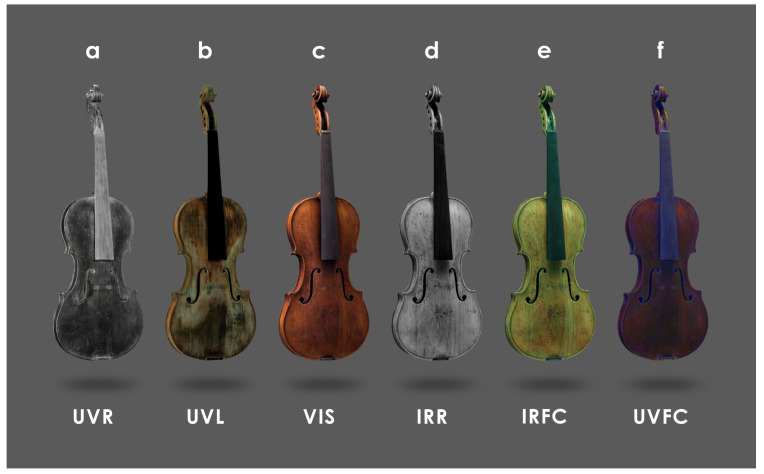
The 3D views of Inv. 107 from the model inspector panel of Sketchfab, represented in different spectral bands: (**a**) Ultraviolet Reflectography (UVR); (**b**) Ultraviolet Luminescence (UVL); (**c**) Visible Light (VIS); (**d**) Infrared Reflectography (IRR); (**e**) IR–False Color (IRFC); (**f**) UV–False Color (UVFC).

**Figure 6 sensors-25-03278-f006:**
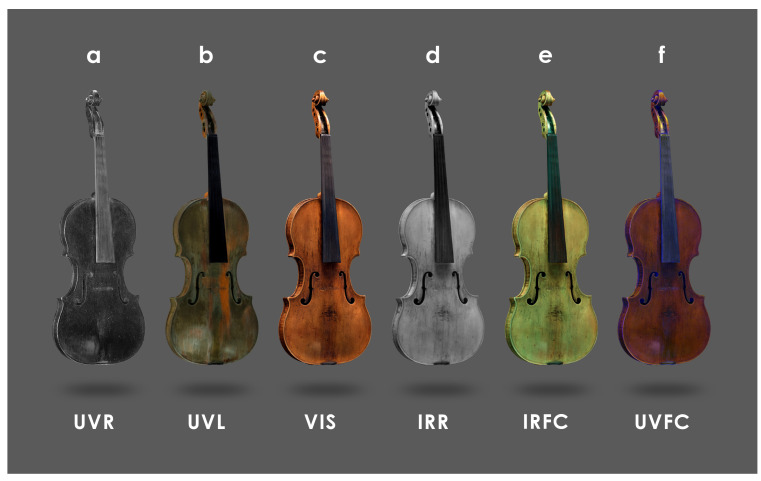
3D views of Inv. 81 from the model inspector panel of Sketchfab, represented in different spectral bands: (**a**) Ultraviolet Reflectography (UVR); (**b**) Ultraviolet Luminescence (UVL); (**c**) Visible Light (VIS); (**d**) Infrared Reflectography (IRR); (**e**) IR–False Color (IRFC); (**f**) UV–False Color (UVFC).

**Figure 7 sensors-25-03278-f007:**
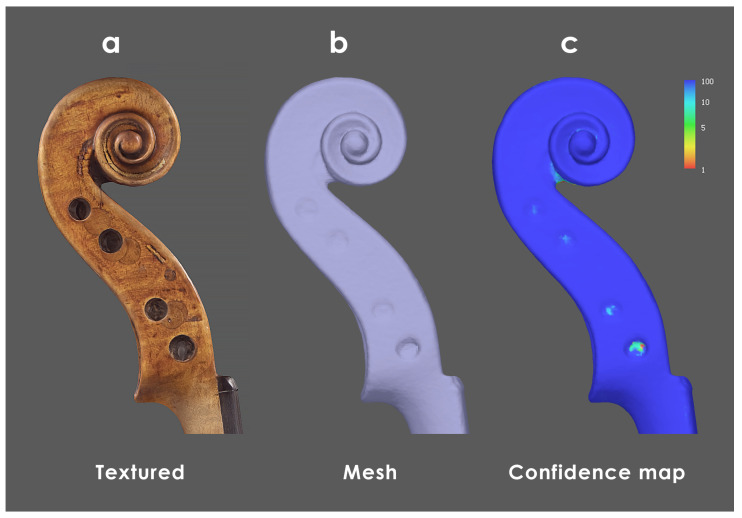
Multiple views of the scroll from Inv. 81. (**a**) VIS textured model; (**b**) polygonal mesh; (**c**) confidence map, where blue and red respectively indicate the best and worst image redundancies. Some particularly hidden details are not adequately recorded. The assessment is performed on the high-resolution models to prevent any artifact induced by mesh decimation.

**Figure 8 sensors-25-03278-f008:**
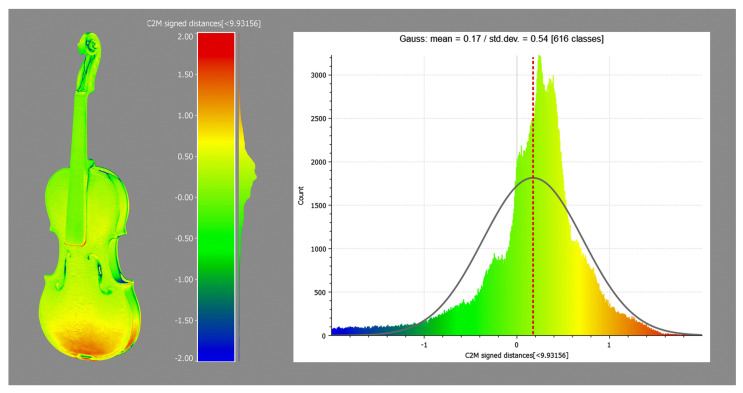
Root Mean Square Error and Standard Deviation for Inv. 107 derived from the comparison between photogrammetry and structured light scanner. Red and blue indicate over- and under-discrepancies, respectively.

**Figure 9 sensors-25-03278-f009:**
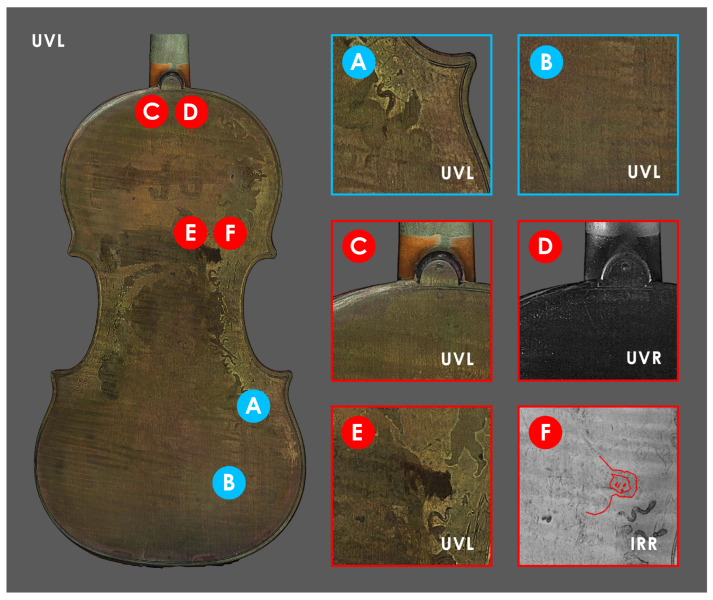
ROIs selected on the back plate of Inv. 81. The ROIs (**A**,**B**) analyzed through ER–FTIR are shown in blue, while in red are the ROIs where various phenomena were observed: (**C**) changes in chromaticity in the UVL and (**D**) absorption in the UVR of the retouching varnish applied around the button and neck heel, (**E**) poor legibility of the face of the figure in the UVL, and (**F**) an increase in the legibility of the same portion in the IRR (in red).

**Figure 10 sensors-25-03278-f010:**
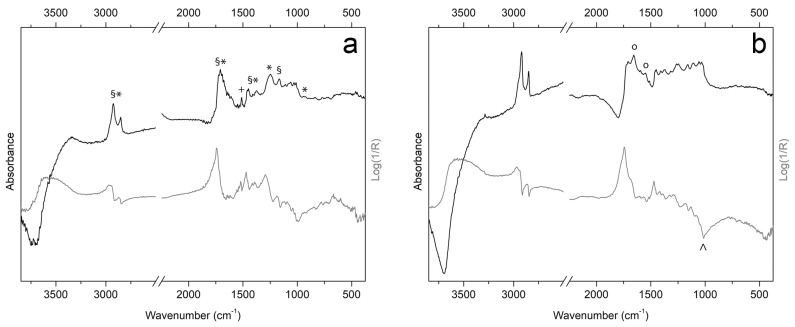
ER–FTIR spectra in pseudo-absorbance (grey line) and after KKT (black line) collected on the back plate of Inv. 81 in correspondence of ROI–A (**a**) and ROI–B (**b**). Marker bands of oil-based varnish (§), shellac resin (*), benzoin resin (+), proteins (o), and talc (^) are highlighted.

**Figure 11 sensors-25-03278-f011:**
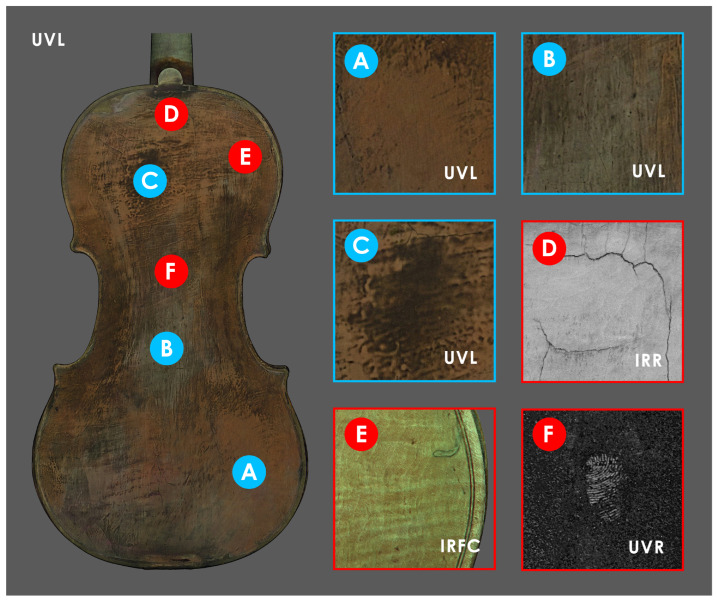
ROIs selected on the back plate of Inv. 107. The ROIs (**A**–**C**) analyzed through ER–FTIR are shown in blue, and in red are the ROIs where various phenomena are observed: a deep crack in the upper part under the button observed through IRR (**D**), a restored woodworm gallery highlighted by IRFC (**E**), and a fingerprint visible in the UVR (enhanced) (**F**).

**Figure 12 sensors-25-03278-f012:**
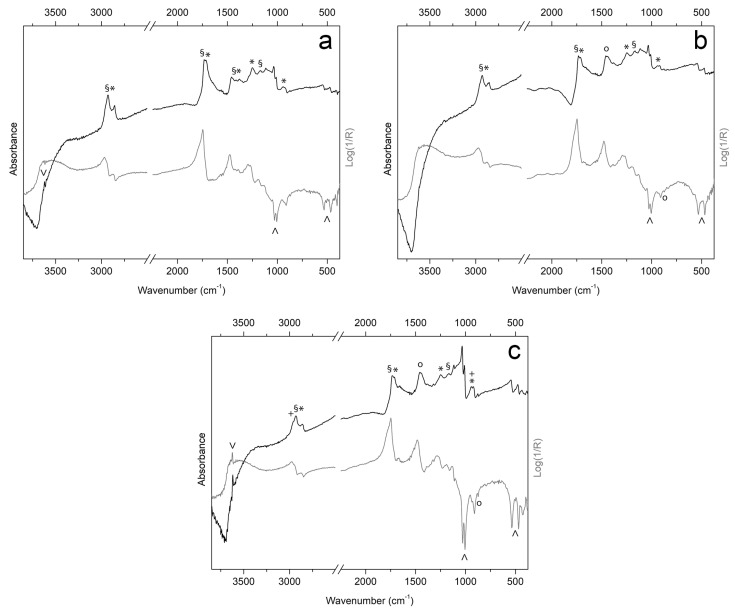
ER–FTIR spectra in pseudo-absorbance (grey line) and after KKT (black line) collected on the back plate of Inv. 107 in correspondence of ROI–A (**a**), ROI–B (**b**), and ROI–C (**c**). Marker bands of oil-based varnish (§), shellac resin (*), long-chain aliphatic compound (+), carbonates (o), and silicates (^) are highlighted.

**Table 1 sensors-25-03278-t001:** Spectral ranges and equipment used for MBI 3D.

Spectral Band	On Camera Filters	Spectral Range	Light Source
UVR	Baader UV	230–390 nm	4 UV LED 365 nm
UVL	Hoya UV–IR Cut + Schott BG40	400–700 nm	4 UV LED 365 nm
VIS	Hoya UV–IR Cut + Schott BG40	400–700 nm	2 Halogen bulbs
IRR	Neewer 800	800–1000 nm	2 Halogen bulbs

**Table 2 sensors-25-03278-t002:** Technical details of the multiband camera used for photogrammetry, and camera setting for Visible Light (VIS), Ultraviolet Reflectography (UVR), Infrared Reflectography (IRR), and Ultraviolet Luminescence (UVL).

FUJI X–T30		Spectral Band	Camera Settings
Sensor	CMOS, 26.1 megapixels	VIS	3 s, f/8, ISO 200
Sensor Size	APS–C, 23.5 × 15.6 mm	UVR	3 s, f/8, ISO 200
Image Size	6240 × 4160	IRR	3 s, f/8, ISO 200
Lens	Minolta MC Rokkor–PF 50 mm f/1.7	UVL	15 s, f/8, ISO 200

**Table 3 sensors-25-03278-t003:** Data from the two photogrammetric models. The difference in image count between the two acquisitions resulted from inconsistent manual rotation of the subject, which introduced variability in the data capture process.

	Inv. 107	Inv. 81
Images (each band)	126	119
Reprojection error	0.499 pix	0.459 pix
Polygons	2,901,226	3,285,132
Texture	8192 × 8192 pix	8192 × 8192 pix

## Data Availability

The raw data supporting the conclusions of this article will be made available by the authors on request.

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
