# Peer review of "Violins Unveiled: A Photogrammetric Framework Integrating Multiband and Spectroscopic Data for In-Depth Examination of Two Musical Instruments"

_sensors, 2025, doi:10.3390/s25113278_

Round 1
Reviewer 1 Report
Comments and Suggestions for Authors
Congratulations for the paper, which is of a high interest. It seems that the integration of Multiband Imaging (MBI) and photogrammetry is opening new doors in research, although it also presents some challenges. It’s great to see how approaches like Physically Based Rendering (PBR) are being used to make 3D models more accessible and understandable, even for those who are not specialists.
The ability to document and analyze materials in historical objects, such as violins, through techniques like spectroscopy is truly fascinating. Additionally, the idea of optimizing the capture of visible light and MBI data to improve efficiency is an exciting step forward. Undoubtedly, the combination of different data sources in architectural and archaeological contexts can revolutionize the way we document and analyze cultural heritage.
The paper is in a great shape, and it could be published as it is. Nevertheless, to reinforce the main targets and goals of the choice of photogrametric MBI I will draft some considerations.
The paper shows a significant application of multiband photogrammetry to two manufactured violins from the 16th and 17th centuries, which are of great cultural interest. Technically, the paper is correct and is of great interest to specialists in the field of technical photography and imaging in cultural heritage applications. Although it is a highly interesting example, perhaps the area that leaves some room for improvement is in the interpretation of the results. The sense and purpose of the article itself may be somewhat blurred, something that could be easily remedied by indicating in that section the reasons for selecting this combination of techniques, with what function and perspective. In a way, it seems that the techniques constitute the end goal of the research rather than the study of the object and the response to specific questions and initial hypotheses. It should be clear what the benefits are of using multiband photogrammetry for the study of cultural artifacts of this nature, such as historical musical instruments. The specific contributions of multiband photogrammetry should be made clear, and why this system is recommended for the study of such objects.
Regarding the conclusions, it is interesting how the integration of Multiband Imaging (MBI) and photogrammetry is highlighted as a promising yet complex area. But it should be highlighted in a deeper way which are the benefits of using such technologies for the study of historial music instruments.
Reviewer 2 Report
Comments and Suggestions for Authors
This article deals about a specific combination of 2D/3D multimodal imaging to enhance visualization and exploration of radiometric features. The first technique is close-range photogrammetry and the second one is multi-band imaging (MBI), also called technical photography in the field of Heritage Science (see CHSOS). The case-study consists of two CH objects (musical instruments). The proposal of the authors is to use PBR rendering feature of Sketchfab to integrate MBI as a multilayer texture, while the title of the paper could make understand merge 4 types of measurements. In my opinion, the title needs to be change to better reflect the approach. The paper is really well-structured, balanced, written and illustrated and making it globally really easy and enjoyable to read. All the technical detail are given, enabling a potential reproducibility of this work. The problem and the research aim is well stated in the introduction. The state of the art is relevant but could be improved to gain in scientific soundness. The results are discussed and the conclusion well addressed.
The paper has overall good quality but could reach excellence level by addressing the following comments.
I declare being expert in the field of CH imaging for HS application, however the spectroscopy section was not evaluated because out of my direct competence and skills.
State of the art Line 51(Major comment):
While the state of the art is complete, some references are out of scope [12 to 26] while others more related to the subject are missing :
- Pioneering work : https://www.isprs.org/proceedings/XXIX/congress/part5/794_XXIX-part5.pdf
- First experiment on radiometric and geometric 2D/3D data registration : https://doi.org/10.1111/j.1477-9730.2011.00664.x
- Multimodal and multi band photogrammetry : 10.5194/isprs-archives-XLVI-2-W1-2022-415-2022
- 3D MBI : 10.1016/j.culher.2018.04.014
- Semantic annotation made on multimodal imaging : 10.5194/isprs-archives-XLVI-2-W1-2022-415-2022
- Integration of non invasive and micro invasive imaging : 10.3390/mps5030052
- Other works dealing with multi band photogrammetry in CH/Hs field :
- https://www.mdpi.com/2571-9408/6/3/148
- 10.2352/issn.2168-3204.2017.1.0.133
- https://doi.org/10.5617/dhnbpub.11094
- https://lirias.kuleuven.be/retrieve/422572
- https://doi.org/10.3390/s23042301
Xtrans sensor Line 123 (Technical detail):
The Fujifilm X–T30 using X-Trans sensor technology (≠bayer matrix), just scientific curiosity does the authors have any references evaluating the use of such unconventional sensor on the image processing, if possible for MBI ?such ref exists for Foveon. In addition, native ISO for X-T30 is 160, even if Xtrans have low noise at noise, it always better to use native ISO (instead of 200 in this case).
Alignment procedure Line 162 (Major comment) :
The alignment (registration for being terminologically more accurate) process has to be better explained, is it truly manual with potential non-linear deformation of the image ? Does it consist of 2D/2D image registration (SIFT like) ? As it implies multispectral imaging, why not Mutual Information or dedicated descriptors (Maximal Self-Dissimilarities) ? This could have been solved or helped by systematizing the data acquisition protocol.
Low poly Line 197 (Minor comment) :
The choice of low-poly must be better explained. Is it because web-based exploration (Sketchfab) ? The discrepancies shown in Fig.8 could come from this decimation. A comparison of low-poly and high-poly (C2M in cloud compare) would give the answer.
3D registration Line 211 (Major comment) :
The authors chose 3D to 3D registration based on ICP instead of other possible methods, this choice is not discussed. Having the ranged-based model, some GCP could have constrained to photogrammetric process to scale and orient the photogrammetric models ? Moreover it could have provided better quality checking by analyzing the uncertainties on the control points and check points. This could have been understandable if Metashape was used in Standard version and not the Professional version, hence it has to be clarified.
Sketchfab Line 217 (Minor comment):
The authors decide to rely on Sketchfab without discussing the aspect of data privacy and sovereignty. This raises many issues and lack of applicabilities for a wide range of CH studies. This is strong limitation of the approach discussed or even mentioned. See Papadopoulos, C.; Gillikin
Schoueri, K.; Schreibman, S. And Now What? Three-Dimensional Scholarship and Infrastructures in the Post-Sketchfab Era. Heritage 2025, 8, 99. https://doi.org/10.3390/heritage8030099
Other alternative would be :
https://dl.acm.org/doi/10.1145/3430846
PBR vs multilayer Line 232 (Major comment) :
The article title needs to be changed has the work is not integrating PBR but rather tweaking a PBR data layout to overlay multispectral imaging into SketchFab. The current title would let reader think that this authors found a method for integrating somehow multi-view and multi-light image collection or any method for capturing optical and physical properties of the material (goniometric measurements, etc). The value of PBR are given but no information is provided on how and why these values. Probably just the minimal ones for not enhancing or altering the texture.
In conclusion, taking into consideration those majors (that need to be revised) and minors (that need to be answered and corrected) I would stand for a major revision for improving the quality. See joint PDF for other potential remarks.

Reviewer 3 Report
Comments and Suggestions for Authors
Overall, this work presents a clear and coherent progression, starting from the fundamentals of photogrammetry and leading to an original and relevant case study on historical violins, while highlighting the added value of integrating multiband data. The well-structured introduction establishes a logical framework, supported by appropriate references, and effectively situates the research within the existing scientific landscape. The research plan is solid, drawing on well-established methods and proposing an innovative approach for the documentation and non-invasive analysis of heritage objects. The description of the instruments, protocols, and software is detailed enough to ensure understanding and, potentially, reproducibility. The results are clearly presented, with well-chosen figures and tables, and the connections made between the different methods strengthen the overall coherence of the narrative. The analysis is rigorous, and the conclusions are directly supported by the data provided. That said, some sections could benefit from further development. For example, a more in-depth discussion of the limitations, concrete examples from cultural heritage and archaeomusicology (Safa 2016, Bellia 2021 & 2022, Zalce 2024), a more thorough justification for the choice of objects and instruments studied, as well as a clearer link between the identified challenges and the methodologies used, could enhance the scientific impact of the article. Likewise, a deeper interpretation of the visual outputs, a discussion of the implications of methodological choices, and broader contextualization with respect to existing research would further enrich the study. Finally, the conclusion could be expanded by emphasizing the concrete contributions of the proposed approach, highlighting its future potential, and more clearly affirming its significance for research in heritage studies and instrumental analysis.
Reviewer 4 Report
Comments and Suggestions for Authors
Please see the attachment.

Round 2
Reviewer 2 Report
Comments and Suggestions for Authors
The authors have considered and revised the manuscript with precision and rigor. All the comments have been addressed either by a revision or a answer, if not both. The manuscript could be accepted in present form.
Just to extend a bit the discussion, the fact that the model itself won't be available to download doesn't solve the issue of using SFab, has the data remains hosted into private server on the company (most probably with at least one back-up in USA). If tomorrow they decide that the model of your Violins could be sold for their own profit through the Fab's marketplace, they possibly could. Just beware.
Reviewer 4 Report
Comments and Suggestions for Authors
The manuscript has been improved compared to the previous version. I have no further suggestions.